# Experimental Evolution of Legume Symbionts: What Have We Learnt?

**DOI:** 10.3390/genes11030339

**Published:** 2020-03-23

**Authors:** Ginaini Grazielli Doin de Moura, Philippe Remigi, Catherine Masson-Boivin, Delphine Capela

**Affiliations:** LIPM, Université de Toulouse, INRAE, CNRS, Castanet-Tolosan 31320, France; ginaini-grazielli.doin-de-moura@inrae.fr (G.G.D.d.M.); philippe.remigi@inrae.fr (P.R.); catherine.masson@inrae.fr (C.M.-B.)

**Keywords:** rhizobia, experimental evolution, nitrogen fixation

## Abstract

Rhizobia, the nitrogen-fixing symbionts of legumes, are polyphyletic bacteria distributed in many alpha- and beta-proteobacterial genera. They likely emerged and diversified through independent horizontal transfers of key symbiotic genes. To replay the evolution of a new rhizobium genus under laboratory conditions, the symbiotic plasmid of *Cupriavidus taiwanensis* was introduced in the plant pathogen *Ralstonia solanacearum*, and the generated proto-rhizobium was submitted to repeated inoculations to the *C. taiwanensis* host, *Mimosa pudica* L. This experiment validated a two-step evolutionary scenario of key symbiotic gene acquisition followed by genome remodeling under plant selection. Nodulation and nodule cell infection were obtained and optimized mainly via the rewiring of regulatory circuits of the recipient bacterium. Symbiotic adaptation was shown to be accelerated by the activity of a mutagenesis cassette conserved in most rhizobia. Investigating mutated genes led us to identify new components of *R. solanacearum* virulence and *C. taiwanensis* symbiosis. Nitrogen fixation was not acquired in our short experiment. However, we showed that post-infection sanctions allowed the increase in frequency of nitrogen-fixing variants among a non-fixing population in the *M. pudica–C. taiwanensis* system and likely allowed the spread of this trait in natura. Experimental evolution thus provided new insights into rhizobium biology and evolution.

## 1. Introduction

The diversity of current living forms is the result of evolutionary processes that have unfolded over billions of years but are difficult to trace in the absence of fossils. With a few exceptions, where dense sampling of natural populations can be done over short periods [1,2,3], only comparative genomics and phylogeny of extant species allow the prediction of ancestral gene contents and evolutionary scenarios. However, adaptive processes can be analyzed in real time via experimental evolution (EE). EE, the propagation of over tens of thousands of generations of organisms in a controlled environment [4,5,6], allows the investigation of the phenotypic, genomic and molecular adaptation thanks to the analysis of fossil records that are generated and stored during the experiment. When laboratory conditions mimic key conditions of natural environments, EE may allow for the understanding of natural evolutionary processes. 

Nitrogen-fixing symbioses have evolved between polyphyletic soil bacteria, called rhizobia, and legumes. In the presence of compatible host plants, rhizobia elicit the formation of novel plant organs, the nodules, which they colonize massively. Inside nodule cells, bacteria reduce large amounts of nitrogen gas into ammonium, which is directly assimilated by the plant and contributes to its growth. On the plant side, this trait is predicted to result from a single gain before the radiation of the Fabales, Fagales, Cucurbitales and Rosales clade (FaFaCuRo), which is followed by multiple independent losses in most descendant lineages [7,8], but maintained in most genera of the Fabales (legumes). On the bacterial side, the trait likely emerged in *Frankia* [9,10]. The genes that determine the synthesis of rhizobial lipo-chitooligosaccharidic Nod factors triggering nodulation are thought to have been transferred from some *Frankia* to a diazotrophic proteobacterium, generating the first rhizobium. After this, nodulation (*nod*) genes and nitrogen fixation (*nif-fix*) genes cluster together on mobile genetic elements, either plasmids or genomic islands, and spread to hundreds of species of distantly related taxa among alpha- and beta-proteobacteria via horizontal gene transfers (HGT) [11,12,13]. Some rhizobia, e.g., the rhizobial *Burkholderia* species, are ancient symbionts of legumes, which probably appeared shortly after legumes emerged, ca. 60 million years ago [14,15,16]. Others, such as *Cupriavidus taiwanensis*, the symbiont of *Mimosa pudica,* evolved more recently. *C. taiwanensis* was estimated to have emerged 12–16 million years ago following a single acquisition of a symbiotic plasmid, likely from *Burkholderia* [15,17,18]. The ancestral rhizobial *C. taiwanensis* then further diversified into five distinct genospecies with a vertical inheritance of the symbiotic plasmid in most cases (Figure 1). 

The *Cupriavidus* branch is a neighbour to *Ralstonia*, a genus that only contains saprophytic and pathogenic strains. The introduction of the symbiotic plasmid of *C. taiwanensis* LMG19424 into the plant pathogenic *Ralstonia solanacearum* GMI1000 strain generated a still pathogenic strain unable to nodulate *Mimosa pudica*, indicating that the acquisition of essential symbiotic genes may not be sufficient to convert a soil bacterium into a legume symbiont. To replay the emergence of a novel genus of rhizobium, here *Ralstonia*, we experimentally evolved this chimeric strain using laboratory conditions reproducing environmental conditions that may have favoured symbiosis [19,20]. Bacteria were repeatedly inoculated onto *Mimosa* plantlets in the absence of other rhizobial competitors (see Figure 2 for the experimental design). At the end of the experiment, 18 final evolved clones were isolated and their evolution analyzed (Figure 1). Despite strong differences between natural and experimental evolution of *Mimosa* symbionts in terms of genetic backgrounds, time frame and environmental conditions (Figure 1), the laboratory process exhibited striking parallels with natural evolution [17]. In particular, adaptation was characterized by a predominance of purifying selection (a purge of non-synonymous mutations) and associated with positive selection in a set of genes that led to the co-option of the same quorum-sensing system in both processes, while no adaptation was observed in the plasmid carrying the genes responsible for the ecological transition. Moreover, this experiment provided new insights into the evolutionary mechanisms leading to endosymbiosis with legumes and on the biology of rhizobia. Here we review some of the lessons learnt from our evolution experiment.

## 2. Evidence for a Two-Step Evolutionary Scenario

The pioneering works of Sullivan et al. demonstrated for the first time the conversion of soil bacteria into legume symbionts in natura through the horizontal acquisition of a symbiotic genomic island [25,26]. Since then, it has been well accepted that rhizobia diversified from independent events of horizontal transfer of essential *nod* and *nif-fix* genes. This finding was then supported by many genomic and phylogenetic analyses [11,15,27,28] and other field experiments [29,30]. However, the transferred genes are not the only ones involved in symbiosis with legumes. Many genes located outside the mobile symbiotic regions have been described as engaged in symbiosis. Among them, specific surface polysaccharide biosynthesis genes (*eps*, *kps*, *lps*, *ndv*) [31,32], metabolism genes (*glnBK*, *ilvCD*, *hemAH*, *leuCB*, *phbAC*, *dme, pckA*) [33,34,35], transporters such as *mdt*, *znu* (zinc)*, dctA* (dicarboxylic acid), and *pstABC* (phosphate) genes [36,37,38,39], chaperones (*groEL*/*groES*) [40] and secretion systems [41] appear to be crucial for symbiosis in different rhizobia. Contrary to the *nod* and *nif-fix* genes, the majority of these symbiotic genes is lineage-specific [33,42]. Using in planta transposon insertion sequencing (Tn-Seq) approaches, Pobigaylo et al. and Flores-Tinoco et al. estimated that 8–15% of *Sinorhizobium meliloti* genes (500 to 900 genes) are involved in symbiosis [43,44]. These genes are scattered throughout the entire genome and mostly involved in metabolism, gene regulation and other cellular processes, confirming the diversity of genes required to achieve symbiosis. On the other hand, some bacterial functions can be deleterious for symbiosis. For example, type 3 secretion systems (T3SS) were shown to decrease the symbiotic capacity of some rhizobium strains or prevent their interaction with some host plants [41,45,46,47]. Furthermore, transfers of symbiotic genes performed under laboratory conditions did not always turn a non-symbiotic organism into an efficient legume symbiont. Indeed, the transfer of diverse rhizobium symbiotic plasmids into *Agrobacterium tumefaciens* or *Escherichia coli,* led to strains that were able to form either pseudo- or non-functional nodules [48,49,50] or nodules fixing low amounts of nitrogen [51]. In line with this, suboptimal symbionts, forming non-fixing nodules, have been observed in the wild following the horizontal acquisition of a symbiosis island [29].

We thus made the hypothesis that the evolution of a new rhizobial species/genus required a post-transfer adaptation step allowing the recruitment and/or the inactivation of specific functions of the recipient genome, especially if the transfer occurred between bacteria with different genomic backgrounds and/or lifestyles. We hypothesized that this optimization step occurred via genome remodeling under plant selection pressure. Indeed, legumes control their microsymbionts through a multistep surveillance system. All along the symbiotic process, plants regularly check the symbiotic status of bacteria and allow, or don’t allow, their proliferation [52]. These checkpoints include the recognition of bacterial Nod-factors and expolysaccharides via specific receptors that convert these signals into a plant developmental program [53,54,55,56,57,58,59], and the production and delivery of ammonium to plant cells, which prevent plant sanctions [60,61,62]. 

We experimentally validated this two-step evolutionary scenario (Figure 2). First, the transfer of the symbiotic plasmid of the *Mimosa* symbiont *C. taiwanensis* into the plant pathogen *Ralstonia solanacearum* led to a non-nodulating (Nod^-^) strain, showing that, in this case, the transfer of the main symbiotic genes was not sufficient to convert *Ralstonia* into *Mimosa* symbionts. Further evolution of this Nod^-^
*Ralstonia* chimeric strain through repeated inoculations to *Mimosa pudica* plantlets and re-isolation of bacteria from nodules allowed the parallel acquisition and progressive improvement of two symbiotic traits (nodulation and nodule cell infection) in independent lineages [19,20,21]. After 16 such cycles, evolved strains of most lineages induced nodules that display features of true nodules: a peripheral vascular system, the production of leghemoglobin, and cells filled with symbiosomes containing bacteria. However, mutualism was not achieved at that stage of the experiment and intracellular bacteria did not persist and prematurely degenerated. It should be noted that in nature rhizobia having suboptimal capabilities of nodulation competitiveness, infection, or nitrogen fixation, or inducing visible plant defense reactions in nodules, have been isolated [29,63,64]. As suggested by our evolution experiment, these natural isolates might correspond to intermediate evolutionary stages before the achievement of symbiosis with their host plant. 

When legumes emerged, around 60–100 million years ago [14,65], nearly all extant proteobacterial genera existed and the different rhizobial lineages had already diverged [66,67]. Among α and β proteobacteria, only a few genera (15 α- and 3 β-proteobacterium genera) have been successfully colonized by symbiotic nitrogen fixation traits, suggesting that only these genera were predisposed for symbiosis. In a comparative genomics study, Garrido-Oter et al. proposed that an ancestral state of adaptation to the root environment in the Rhizobiales predisposed these bacteria to symbiosis [68]. Based on a Transposon-Sequencing (Tn-Seq) approach, Salas et al. also came to the same conclusion that rhizobia were adapted to the rhizosphere of their host plants before their specialization in nodulation [69]. The adaptation to rhizosphere colonization may include the capacity to use specific plant metabolites [70,71] and tolerance to biotic and abiotic stresses. Predisposition to legume symbiosis may require, in addition, the production of specific surface polysaccharides (EPS, KPS, LPS), and the capacity to avoid plant defenses [72,73]. Although we did not identify which genetic features predispose a bacterium to legume endosymbiosis, we provided evidence that a strain with a completely different lifestyle (pathogenic and strictly extracellular) can be converted into an intracellular legume symbiont. This revealed that indigenous functions of the recipient genome were recruited and/or suppressed for symbiosis. The capacity of *R. solanacearum* to thrive in many plant environments and infect more than 250 different plant species [74], including some legume species [75], might have contributed to its predisposition to symbiosis.

Phylogenetic studies predicted that mutualistic bacteria often derived from pathogens whilst the opposite was very rare [76]. *C. taiwanensis* might indeed have evolved from an opportunistic pathogen, since the closest identified bacterium to symbiotic *C. taiwanensis* was isolated from a cystic fibrosis patient (*C.* sp. LMG19464) [77], and these symbionts have an atypical rhizobial T3SS, which is not connected to the regulation of nodulation [45] and whose organization is similar to the T3SS of the human opportunistic bacterium *Burkholderia cenocepacia* [78]. Transitions from parasitism to mutualism were empirically demonstrated in other evolution experiments, that, for instance, converted *E. coli* parasitic phages [79], nematode pathogenic bacteria [80] or mice gut fungal pathogens [81] into beneficial symbionts.

## 3. Regulatory Rewiring of the Recipient Genome as a Main Driver of Symbiotic Adaptation

While the genetic mechanisms of transfer of mobile genetic elements, including symbiosis plasmids and genomic islands, have been widely studied and well described [82,83,84,85], post-transfer adaptation steps are still underexplored. Our evolution experiment allowed us to tackle this issue. To understand how *R. solanacearum* that acquired a symbiotic plasmid further evolved into *Mimosa* intracellular symbionts, we resequenced the genome of many intermediate and final evolved clones along parallel lineages. This led us to analyze the genotype/phenotype correlations and identify the main symbiosis-adaptive mutations. 

The capacity to enter roots and form nodules was unlocked by the inactivation of the T3SS (*hrcV* mutation), which is the main virulence factor of *R. solanacearum* [74]. The sole inactivation of the T3SS led to the formation of necrotic nodules that were only extracellularly invaded. A first level of intracellular infection, i.e., infection of nodule cells, was gained via the inactivation of virulence regulators, either VsrA (frameshift mutation in the *vsrA* gene) or HrpG (stop mutations in the *hrpG* gene, or mutation in the *hrpG* promoter region or in the sigma factor *prhI* acting upstream from HrpG in the regulatory cascade) [21,22]. The stop mutations in HrpG allowed the concomitant gain of the nodulation capacity since HrpG controls the T3SS and associated effectors [86], while other mutations (in *vsrA*, *hrpG* promoter and *prhI*) arose after the *hrcV* mutation having unlocked nodulation (Figure 2B). HrpG and VsrA thus likely control functions that positively or negatively interfere with intracellular infection. A small number of infected cells and the presence of necrotic zones characterized this first level of intracellular infection. A second level of infection, characterized by a high number of infected cells and an almost total absence of necrosis, was then reached in several lines. This second level of infection was obtained via mutations affecting global transcription regulators, either EfpR or PhcA. Missense mutations in *efpR* itself or an intergenic mutation upstream from a gene of unknown function triggered a constitutive repression of EfpR [23], while mutations in two different components of the Phc quorum sensing system (*phcB* and *phcQ*) led to a modulation of the quorum sensing threshold activating PhcA [24]. EfpR and PhcA were shown to act as both central players of the *R. solanacearum* virulence regulatory network and global catabolic repressors [23,87,88,89]. Interestingly, two genes of the *phc* regulatory pathway, *phcB* and *phcS*, display a signature of positive selection in naturally evolved *Mimosa* symbionts and we showed that PhcA was co-opted for symbiosis in *C. taiwanensis* LMG19424 [17].

Altogether, our analysis showed that the rewiring of regulatory circuits was the main driver of the symbiotic adaptation of *Ralstonia*. Strikingly, all identified adaptive mutations for infection affect global regulatory proteins (HrpG, VsrA, EfpR, PhcA) controlling, positively or negatively, the expression of several hundreds of genes (Figure 3) [23,86,89,90,91,92]. This is in accordance with other evolution experiments in which global regulators were frequently targeted [93,94,95,96,97]. Our results confirmed that mutations in global regulators can be highly beneficial by changing the expression of hundreds of genes in a concerted way, particularly during the first steps of adaptation to a new environment [98]. The fact that the conversion of *R. solanacearum* into intracellular legume symbionts was mainly driven by regulation rewiring means that ancillary functions required for nodule infection are there but need to be expressed at the right time and place. A future challenge will be to identify which functions controlled by the targeted regulators are beneficial or deleterious for intracellular infection in the *Ralstonia*–*Mimosa* interaction. We hypothesize that the same functions are affected in the different lines, but the large overlap between the EfpR and PhcA regulons [24] makes this task difficult (Figure 3).

In natural rhizobia, a large number of regulatory systems involved in symbiosis were described. These systems vary a lot among rhizobia, even in close natural taxons. For example, the expression of nodulation genes is regulated by NodD proteins in all rhizobia. However, depending on rhizobium species, the expression of these genes is controlled by either a single NodD regulatory protein [78] or multiple, both positive and negative, regulators organized in intricate networks [99,100,101,102,103]. Likewise, expression of nitrogen fixation (*nif*-*fix*) genes is under the control of the conserved NifA protein, but upstream from NifA the regulatory cascade is highly variable among rhizobia, involving different two-component systems (FixLJ, RegSR, NtrBC, ActRS, FixL-FxkR) responding to various signals [104,105,106,107] and specific downstream transcription regulators (FixK1, FixK2, FnrN) [108,109,110]. Surface polysaccharide biosynthesis genes are other striking examples of essential symbiotic genes subjected to very complex regulations involving a multitude of interconnected regulators that largely differ between rhizobium species [111,112]. Integration of symbiotic functions (both endogenously recruited and horizontally acquired) in the regulatory circuits of the recipient bacterium is thus probably a key step in the evolution of new rhizobium species. In our EE, depending on the lineages, different regulatory pathways were modified by bacteria to achieve nodulation and infection, thus making a noticeable parallel between natural and experimental evolutionary processes of rhizobia.

So far, no adaptive mutations were identified on the symbiotic plasmid in the experiment, indicating that adaptation mainly occurred on the recipient genome. This is in accordance with the lack of evidence of positive selection of plasmidic genes in natural *Mimosa pudica* symbionts [17]. A possible explanation is that in the natural evolution, like in the experimental one, the symbiotic plasmid originated from a more ancient *Mimosa* symbiont [13,16], and did not require further adaptation to the legume host.

## 4. The Legume Plant, a Strong Selection Pressure for Shaping Bacterial Endosymbiotic Evolution

Horizontal gene transfer (HGT) of symbiotic genes is frequent in nature [113,114], which is consistent with the observations that rhizosphere and nodule environments promote HGT [85,115]. Given that rhizobia co-exist in the soil, and even in nodules, with many other different bacteria [116,117], the transfer of symbiosis genes to new bacterial genera must be extremely common. The fact that rhizobia are restricted to a relatively small number of bacterial genera [13,118] suggests that exceptional circumstances are needed for the successful actualization of symbiotic potential in strains having received the *nod* and *nif-fix* genes. We can infer what these circumstances are by learning from field experiments with Lotus, *Biserrula pelecinus* and soybean symbionts [23,27,28,116]: (i) introduction of an exotic legume plant with a compatible rhizobium which is mal-adapted to the new environmental conditions, (ii) the presence of endogenous soil bacteria already adapted to the environment where the legume is introduced, and (iii) the absence of competing rhizobia.

Our experiment shows that, once these circumstances are reproduced in the laboratory, the plant exerts a strong selection pressure for the full expression and optimization of symbiotic traits. It was already well known that the plant selects the most nodulation-competitive strains. While large bacterial populations are present in the rhizosphere [119,120,121], each nodule is formed by one or a few bacteria [122]. Thus, bacterial root entry and nodulation represent a strong bottleneck and, in a mixed community, only the most competitive bacteria successfully infect the plant [123]. This property was used as a genetic screen in the past [124] and was evidenced in our EE where nodulation competitiveness strongly improved all along cycles [19]. In most cases, the genetic factors underpinning this phenotype have not been identified in our experiment (with a few exceptions, see below). Adaptive mutations improving nodulation could affect genes involved in diverse functions (growth in the rhizosphere, motility, biosynthesis of cell surface components) since it is known that multiple bacterial traits affect nodulation competitiveness [123,125]. 

Our evolution experiment showed for the first time that a late symbiotic stage, the infection of nodule cells, could be acquired and further improved by plant-mediated selection through two mechanisms. Firstly, bacteria multiply at much higher densities during intracellular infection (ca. 10^8^–10^9^ bacterial cells per nodule) than in infection threads and extracellular spaces (ca. 10^6^ bacteria per nodule). Secondly, we observed a coupling between infection and nodulation since all mutations that improved intracellular infection also improved nodulation competitiveness [22] (and unpublished data). The genetic link between infection and nodulation is supported by the finding that mutations optimizing infection in our EE modify the expression of bacterial genes during the earliest stages of symbiosis (infected root hairs) [24]. This coupling is also consistent with the known effect of Nod factors on intracellular infection [126,127,128] and the fact that the intracellular release of rhizobia in primodium cells depends on the proper progression of infection threads [129], which otherwise conditions nodule development. From an evolutionary perspective, the nodulation/cell invasion coupling enables the selection at the root entry level of a bacterial trait that is only expressed at the later stages of the symbiotic process. These results also call for a more extensive examination of this question by systematically measuring nodulation competitiveness of non-infectious rhizobial mutants, such as exopolysaccharide deficient mutants, which has been never, or at best extremely rarely, performed. 

However, the acquisition of intracellular infection was not universal in our EE as some lines (ancestor CBM356 and derived lineages with 42-day cycles) did not evolve infection (Figure 2B). Thus, depending on chance (i.e., the nature of the first nodulation-adaptive mutation) and selective regime (long (42 day) vs. short (21 day) cycles), extracellular-infecting clones may prevail over intracellular ones, possibly leading to an evolutionary dead-end.

Following their accommodation into plant cells and when oxygen concentration is low enough, rhizobia initiate nitrogen fixation, the critical process required for nutrient exchanges during this mutualistic symbiosis. A number of studies have shown that nitrogen fixation is not selected at the root entry/nodulation level per se [60,130,131,132], unless it is associated with another trait [133,134,135]. Instead, post-infection control mechanisms promote fitness of fixing bacteria over non-fixing ones [61,62]. These mechanisms were hypothesized to contribute to the stability of mutualism over long time-scales [136,137], by limiting the proliferation of non-fixing bacteria that may otherwise outcompete fixing bacteria that invest cellular resources into the energetically costly process of nitrogen fixation. In our system, we were interested in tackling another question: do post-infection control mechanisms allow the evolution of mutualism in natura and in our experimental setup? In other words, would nitrogen-fixing clones be selected in our EE, if they happen to arise? Using the reference *C. taiwanensis/M. pudica* system and isogenic Fix^+^ and Fix^-^ bacterial mutants, we followed the dynamics of the two bacterial populations along the symbiotic process and identified the timing and extent of the post-infection control mechanisms specifically targeting Fix^-^ cells [60]. We show that Fix^-^ bacteria multiply similarly to Fix^+^ bacteria in nodules, but only until 16–21 days post-infection, the time at which their fitness begins to decline compared to Fix^+^ bacteria due to a premature degeneration. Interestingly, differential survival of Fix^-^ vs. Fix^+^ bacteria occurred when single nodules were co-infected by the two bacterial genotypes. This, together with similar results obtained with *Lotus* and *Acmispon* [138], indicated the existence of cell-autonomous sanctions. Moreover, mathematical modeling predicted the impact of different environmental factors on the kinetics of invasion of rare Fix^+^ clones in a Fix^-^ population, and these predictions were experimentally validated by witnessing the rapid increase in frequency of Fix+ bacteria along serial inoculation–nodulation cycles [60]. An important topic for future research will be to identify the molecular bases of post-infection control mechanisms. Previous work showed that soybean nodules could restrict oxygen diffusion in non-fixing nodules [61], but the generality and mechanistic details of this process remain to be investigated.

Together, our work has shown that plant-mediated selection mechanisms were efficient in promoting the successive evolution of key symbiotic properties: nodulation, intracellular infection and nitrogen fixation. While we focused here on plant-mediated selection factors observed in our EE, it is worth remembering that additional biotic and abiotic factors (including physical environment, soil bacterial communities, plant life histories) also shape bacterial adaptation in natural populations [139].

## 5. Discovery of a Hypermutagenesis Mechanism that Accelerates HGT-Based Evolution

The generation of genetic variation through de novo mutations fuels bacterial adaptation to new environments. Mutations in genes involved in DNA repair, such as *mutS/L/H*, can lead to a hypermutator phenotype, characterized by a constitutive elevation of mutation rate and, consequently, increased evolvability. Hypermutator genotypes are often selected in evolution experiments or during latent host colonization [140,141]. Bacteria can also increase mutation rates transiently in response to stress, a mechanism known as stress-induced mutagenesis (SIM) [142,143]. SIM is due to the expression of error-prone DNA polymerases under the control of major stress-induced regulatory pathways such as SOS or RpoS. Despite the long-term recognition of this phenomenon, the contribution of SIM to bacterial adaptive evolution has received relatively little experimental scrutiny.

Resequencing evolved clones revealed a large number of mutations in our experiment, although the strains were not constitutive hypermutators. Additional sequencing and experiments revealed that bacterial clones underwent transient hypermutagenesis (~5–10× increase in mutation rate) during each exposure to the plant culture medium, with a further increase (~4×) in the presence of *Mimosa pudica* plantlets. We identified error-prone DNA polymerases from the *imuABC* family that were located on pRalta and responsible for this transient hypermutagenesis [144]. In our system, SIM accelerates adaptation since, when re-evolved for 5 additional cycles, clones carrying *imuABC* genes (like our original ancestor) adapted faster than isogenic *imuABC* mutants evolved in the same conditions [144]. Interestingly, the interrogation of genomic databases revealed than more than half of symbiotic plasmids carried error-prone DNA polymerases. It is tempting to speculate that the co-transfer of symbiosis genes with mutagenesis cassettes may facilitate adaptation to new symbiotic environments following plasmid acquisition by increasing the probability to generate variants with improved symbiotic capabilities that the plant can select. The fact that the natural process, as the experimental one, resulted in rapid genetic diversification dominated by purifying selection, also argues in favor of the role of *imuABC*-based transient mutagenesis in the natural evolution of rhizobia. Whether dependent on plasmidic or chromosomal genes, SIM may contribute to bursts in mutation rates occurring during host-microbe interactions [145,146]. Moreover, it is worth mentioning that other mechanisms can increase evolvability in rhizobia, such as the activation of HGT by root exudates [85], the transposition of insertion sequence elements [47,147], and the Non-Homologous End-Joining mechanism (NHEJ) [148].

Balancing the benefits during the early steps of adaptation to a new environment, hypermutagenesis can compromise fitness on the long term due to the accumulation of deleterious mutations [140]. This can lead to selection against hypermutagenesis, consistent with the modulation of mutation rates observed in several evolution experiments [149,150,151]. Testing whether SIM can be counter-selected would bring interesting new arguments into the debate over the adaptive nature of SIM. In that respect, it is intriguing to note that SIM does not occur in the original host of pRalta, *C. taiwanensis*, but the causes, environmental or genetic, are unknown.

## 6. New Insights into the Biology of Evolutionary Protagonists

EE uses extant strains with specific properties or lifestyles as protagonists. The tracking of mutations underlying phenotypic changes during laboratory evolution may reveal unidentified molecular players of functions otherwise studied by traditional molecular genetics [152,153]. In our experiment, this potential to advance the understanding of the biology of ancestral strains was highlighted by the discovery of new virulence and symbiotic components in *R. solanacearum* and *C. taiwanensis*, respectively.

The virulence mechanisms of *R. solanacearum* have been genetically dissected for many years. This allowed for the identification of a complex regulatory network involving many interconnected regulatory cascades, which controls the different functions involved in pathogenicity [154,155,156,157,158]. In this network, the LysR family transcriptional regulator PhcA was identified as a central regulator, whose activation is controlled by the quorum sensing (QS) PhcBSR(*Q*) system [159]. However, the role of the last gene of the operon, *phcQ,* could not be evidenced [160]. In our experiment, we focused on a *phcQ* mutation that enhanced the capacity of bacteria to infect nodule cells. This led us to show that this mutation delayed the cell density-dependent activation of PhcA and further showed that *phcQ* is involved in the production of QS molecules [24]. In addition, our experiment and another one conducted in parallel [161] revealed the occurrence of a virulence pathway previously overlooked, the *efpR* pathway. Mutations in this gene or in one upstream regulatory component allowed a better in planta colonization of *R. solanacearum* evolved on different host plants [87] and an improved intracellular infection capacity of the chimeric *Ralstonia* evolved on the *M. pudica* legume [23]. Mutation in the upstream yet unidentified regulatory component (a SNP in the intergenic region upstream from the Rsc0965 gene) led to a surprising complete loss of virulence. 

The *nod* genes conveyed by symbiotic plasmids and islands determine the biosynthesis of signalling compounds, called Nod factors, which trigger the plant developmental program leading to nodulation and infection in legumes [162,163]. In 2008, Amadou et al. found that the Nod factors produced by *C. taiwanensis* were consistent with the in silico prediction based on the genome sequence and the identified *nod* genes [78]. They were found to be pentameric chito-oligomers sulphated at the reducing end, N-acylated by vaccenic acid or palmitic acid and mostly substituted by an N-methyl and a carbamoyl group at the non-reducing end. This was recently challenged from the analysis of one of our evolved lines. In the R line, the capacity to infect nodule cells was not obtained after 16 cycles of evolution despite the detection of an adaptive mutation for intracellular infection (*hrpG*A179V) in the final R16 clone [20]. Searching for the modification that countered the adaptive effect of the *hrpG*A179V mutation led us to investigate the role of a gene deleted in the symbiotic plasmid of R16 and whose reintroduction restored nodule cell infection capacity [164]. We found that this gene, now called *noeM*, is a host-specificity nodulation gene responsible for a Nod factor modification that was previously missed, the opening and oxidation of the reducing end of the molecule [165]. Surprisingly the inactivation of *noeM* did not affect the nodule cell infection capacity but only the nodulation competitiveness of its natural host *C. taiwanensis* in interaction with *M. pudica*, indicating that its impact on symbiosis depends on the genomic background.

## 7. Conclusions

The evolution of nitrogen-fixing root nodule symbioses was a key innovation that gave access to new ecological niches to both plant (nitrogen-depleted soils) and bacterial (root nodules) symbionts. Comparative genomic and phylogenomic approaches have greatly contributed to understanding the evolution of these complex interactions [7,8]. By using EE on the bacterial partner, we were able to complement these approaches and provide a dynamic view of evolution, although in a somewhat artificial context, since it is impossible to know and reproduce the environmental conditions and protagonists that took part in the emergence of this symbiosis. EE did not address the question of the origin of the first rhizobia, nor that of the co-evolution between plants and rhizobia. Instead, it tackled the issue of the spreading of the N_2_-fixing capacity in many proteobacterial branches. One key lesson was that a bacterium with a completely different lifestyle (pathogenic, extracellular) can be turned into an intracellular legume symbiont via the acquisition of mutualistic genes and regulatory rewiring under plant selection pressure. This result helps us to understand why so many bacterial branches have successfully been colonized by nitrogen-fixing traits. So far, our work focused on the first symbiotic stages: acquisition and improvement of nodulation and nodule invasion. By continuing the evolution experiment and its genetic analysis, future work should address the evolution of late stages of symbiosis, such as the acquisition of persistence, nitrogen fixation and mutualism and should provide a better understanding of these poorly known symbiotic stages. EE including a single selection cycle also proved to be very useful in understanding the bacterial genetic bases of incompatible symbiotic interactions [47,166,167] and to analyze plant-mediated selection acting on rhizobial communities [168,169]. Complementary to synthetic biology and classical genetic approaches, EE will undoubtedly become an indispensable tool to select efficient, host-adapted microsymbionts in the perspective of optimizing current symbiotic associations or transferring N_2_-fixation in crops [170,171].

## Figures and Tables

**Figure 1 genes-11-00339-f001:**
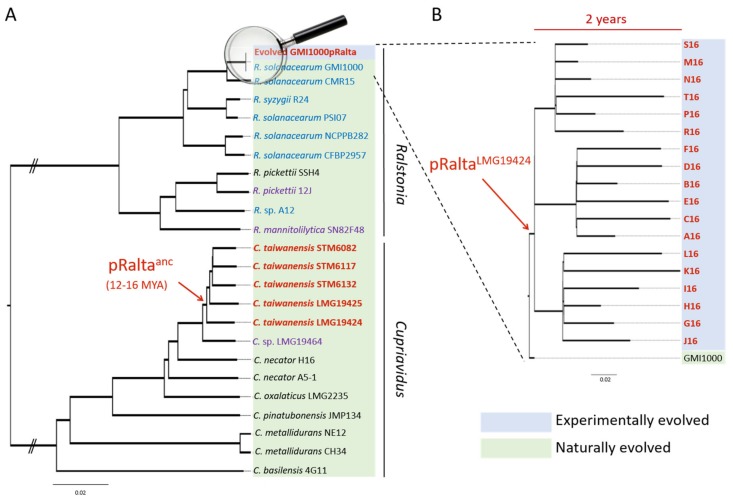
Phylogeny of naturally and experimentally evolved *Mimosa pudica* symbionts in the *Cupriavidus* and *Ralstonia* branches. (**A**) Neighbour-joining phylogeny of selected *Ralstonia* and *Cupriavidus* strains based on 1003 gene families defined as core genome by [17]. Evolved “GMI1000pRalta” represents a group of 18 closely related strains derived via experimental evolution from the strain *Ralstonia solanacearum* GMI1000. (**B**) Close-up of the phylogenetic relationships between the 18 resequenced clones evolved for 16 cycles (two years) compared to the ancestor (*R. solanacearum* GMI1000). This neighbour-joining phylogeny is based on single-nucleotide polymorphisms identified in the experimentally evolved strains. (A) and (B) *M. pudica* symbionts are in red and bold. Plant pathogens, opportunistic human pathogens and saprophytic strains are in blue, purple and black, respectively. Arrows indicate the acquisition of a symbiotic plasmid, either the ancestor of the *Cupriavidus taiwanensis* one (pRalta^anc^) during natural evolution or that of *C. taiwanensis* LMG19424 (pRalta^LMG19424^) during experimental evolution.

**Figure 2 genes-11-00339-f002:**
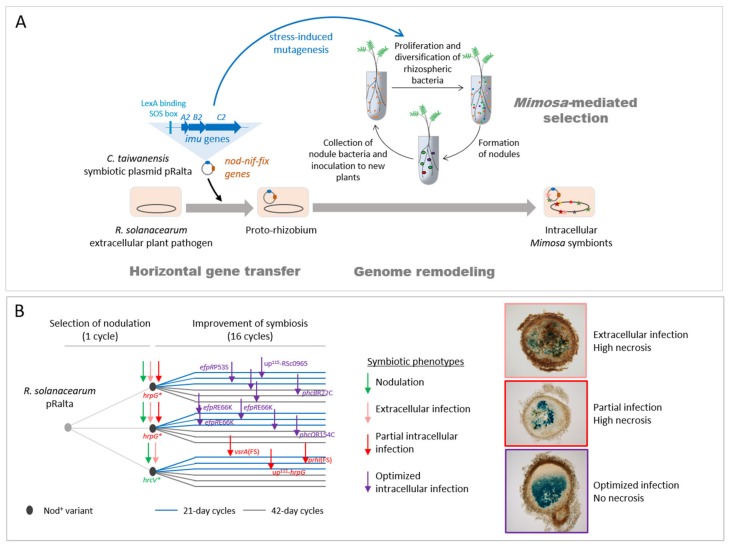
Experimental evolution of the plant pathogen *Ralstonia solanacearum* into legume symbionts. (**A**) The symbiotic plasmid (pRalta) of *Cupriavidus taiwanensis*, the natural symbiont of *Mimosa pudica*, was introduced into *R. solanacearum* [21], generating a chimeric strain unable to nodulate *M. pudica*. This proto-rhizobium was then evolved through serial cycles of inoculation to *M. pudica* plantlets and re-isolation of nodule bacteria. In the first cycle, three nodulating variants were entrapped by the plant, and in the following cycles these variants were symbiotically improved (see panel B). In addition to the *nod* and *nif-fix* genes involved in Nod Factor synthesis and nitrogen fixation respectively, the pRalta plasmid carries a mutagenesis cassette (*imuA2B2C2*) that elevates mutation rate when bacteria are free-living in the plant culture medium. This transient hypermutagenesis increases the genetic diversity of the rhizospheric bacterial population among which the plant selects the most beneficial variants. (**B**) 18 parallel lineages were derived from the three nodulating (Nod^+^) variants via two different selection regimes (cycles of 21 or 42 days). After acquisition of the nodulation ability, bacteria progressively improved their capacity to infect nodules. Three levels of infection were observed in the evolved lines: extracellular, partially intracellular and nicely intracellular. The main adaptive mutations responsible for the acquisition and improvement of symbiosis were identified [21,22,23,24]. *, stop mutation. FS, frameshift. up^112^, up^115^, intergenic mutations located 112 bp and 115 bp upstream from the gene indicated.

**Figure 3 genes-11-00339-f003:**
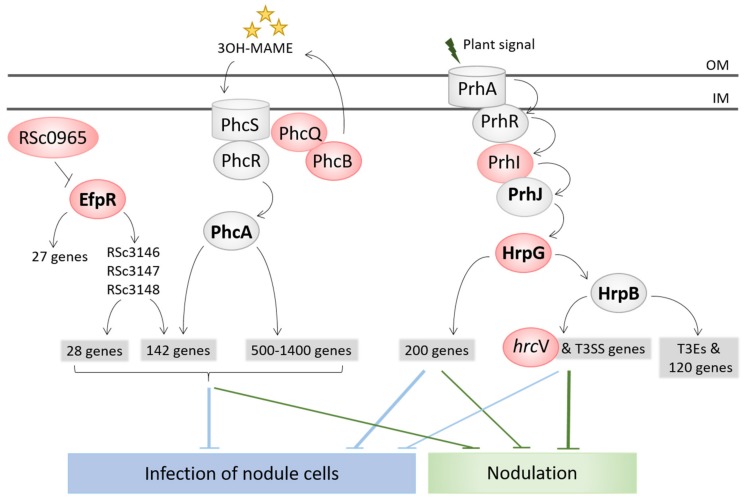
Adaptation to endosymbiosis mainly occurred through regulatory rewiring. Genes mutated in the experiment are indicated in red. Nodulation was gained by inactivation of the T3SS via a stop mutation in *hrcV* [21]. A first level of intracellular infection was obtained by the inactivation of the virulence regulator HrpG [21,22]. Optimized intracellular infection was reached either by inactivation of the EfpR pathway or a modulation of the quorum sensing pathway controlling PhcA [23,24]. EfpR and PhcA controls a common set of genes as well as other specific genes. It is not known which group of genes controlled by EfpR and/or PhcA interfere with intracellular infection. Transcriptional regulators are in bold. 3OH-MAME, (*R*)-3-hydroxymyristic acid methyl ester, quorum sensing molecules produced by the GMI1000 strain of *R. solanacearum*.

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
