# Peer review of "Experimental Evolution of Legume Symbionts: What Have We Learnt?"

_genes, 2020, doi:10.3390/genes11030339_

Round 1
Reviewer 1 Report
I read the submitted review entitled "Experimental evolution of legume symbionts: what have we learnt?"
I think it is a good review and have significant data that will support the researchers in the field of agronomy.
I recommend publication of this review with minors revisions especially in English that have some typographical errors
Author Response
- I recommend publication of this review with minors revisions especially in English that have some typographical errors
> The text has been carefully read by a native English person. Spelling and grammar were also checked automatically in Microsoft Word.
Reviewer 2 Report
This review report is supposed to elucidate the natural evolutional mechanism of Rhizobium in relation to the chimeric strain and symbionts created in sequence at the short-term horizontal transfer in the lab. The gist of the study is written in clear English and understandable and it was really interesting to read. But I noticed some points as follows to make the report more contributive.
- The numbering of the paragraphs is incorrect.
- Detailed explanation of the endosymbiosis of Rhizobium and host plants (Fabaceae) is needed to be explained earlier to make the report more persuasive. (especially about the plant selection pressure in L127-132 is preferred to be summarized and added in the introduction)
- The points of similarity of the original lab experiments and natural evolution are needed to be mentioned clearly. (L75-76)
- The relation of the increase of the gene diversity of the original lab experiments and its role in the natural biological evolution is a little weak.

Author Response
- The numbering of the paragraphs is incorrect.
> This was corrected (Line 428).
- Detailed explanation of the endosymbionts of Rhizobium and host plants (Fabaceae)is needed to be explained earlier in the introduction to make the report more persuasive. (especially about the plant selection pressure in L127-132 is preferred to be summarized and added in the introduction).
> The objective of the review was to present and comment the different lessons learnt from our experimental evolution. For each lesson, we explained the context and hypotheses made at the time we started the experiment and discussed the lesson in the light of our knowledge on rhizobia and their natural evolution. For this reason, we prefer to keep the discussion about plant selection in the corresponding paragraph “Evidence for a two-step evolutionary scenario”, to be homogenous with the remaining of the review.
>Yet to better expand the introduction, we moved the following sentences (from the conclusion) to the introduction (Lines 42-51): “On the plant side, this trait is predicted to result from a single gain before the radiation of the Fabales, Fagales, Cucurbitales and Rosales clade (FaFaCuRo), which was followed by multiple independent losses in most descendant lineages [164, 165], but maintained in most genera of the Fabales (legumes). On the bacterial side, the trait likely emerged in Frankia [9, 10]. The genes that determine the synthesis of rhizobial lipo-chitooligosaccharidic Nod factors triggering nodulation are thought to have been transferred from some Frankia to a diazotrophic proteobacterium generating the first rhizobium. Then nodulation (nod) genes and nitrogen fixation (nif-fix) genes have clustered together on mobile genetic elements, either plasmids or genomic islands, and spread to hundreds of species in distantly related taxa among alpha- and beta-proteobacteria via horizontal gene transfers (HGT) [11, 12, 13].”
- The point of similarity of the original lab experiments and natural evolution are needed to be mentioned clearly. (L75-76)
> We added the sentence (Lines 90-94) “Specifically, adaptation was characterized by a predominance of purifying selection (a purge of non-synonymous mutations) and associated with positive selection in a set of genes that led to the co-option of the same quorum-sensing system in both processes, while no adaptation was observed in the plasmid carrying the genes responsible for the ecological transition.”
- The relation of the increase of the gene diversity of the original lab experiments and its role in the natural biological evolution is a little weak.
> In Remigi et al. 2014 we demonstrated that the increase in diversity due to the activity of the imuABC cassette accelerated the experimental evolution of symbiotic Ralstonia. The fact that more than half of symbiotic plasmids of natural rhizobia carried error-prone DNA polymerases suggests that the co-transfer of symbiosis genes with mutagenesis cassettes facilitated adaptation to new symbiotic environments following plasmid acquisition in natura. The fact that the natural process, as the experimental process, resulted in rapid genetic diversification dominated by purifying selection, is another argument in favor of the role of the plasmid mutagenesis cassettes in the natural evolution of rhizobia.
> We thus added this last argument in the text (Lines 371-375): “It is tempting to speculate that the co-transfer of symbiosis genes with mutagenesis cassettes may facilitate adaptation to new symbiotic environments following plasmid acquisition by increasing the probability to generate variants with improved symbiotic capabilities that the plant can select. The fact that the natural process, as the experimental one, resulted in rapid genetic diversification dominated by purifying selection, also argues in favor of the role of imuABC-based transient mutagenesis in the natural evolution of rhizobia”.
Reviewer 3 Report
This is a very well written review, that summarizes the authors' published work on experimental evolution of nodulation (in the Ralstonia background), and places it nicely in the context of rhizobial biology and evolution. I have very little to criticize about the paper, except a few small grammar issues here and there.
Some minor comments:
LINE 32. Scenarios is the more common and accepted plural. Scenarii is rare and nonstandard.
Fig 1 I found quite confusing. I suggest splitting in two to portray natural evolution and experimental evolution in different figures.
Line 120. Ref 47 did actually show N fixation in the Agrobacterium transconjugants, on beans.
Author Response
- LINE 32. Scenarios is the more common and accepted plural. Scenarii is rare and nonstandard.
> Done
- Fig 1 I found quite confusing. I suggest splitting in two to portray natural evolution and experimental evolution in different figures.
> The idea of this figure was to show the phylogenetic relationships between naturally and experimentally evolved Mimosa symbionts as well as the respective amplitude of their evolution. Whatever their origin (soil, plant, animals), their lifestyle (saprophytic, pathogen, mutualist) or their evolution conditions (experimental, natural), a phylogeny between existing strains can be established.
> Lines 65-75 : To clarify this we changed the title and legend of figure 1 into “Figure 1. Phylogeny of naturally and experimentally evolved Mimosa pudica symbionts in the Cupriavidus and Ralstonia branches. (A) Neighbour-joining phylogeny of selected Ralstonia and Cupriavidus strains based on 1003 gene families defined as core genome by [15]. Evolved “GMI1000pRalta” represents a group of 18 closely related strains derived via experimental evolution from the strain R. solanacearum GMI1000. (B) Close-up of the phylogenetic relationships between the 18 resequenced clones evolved for 16 cycles (2 years) compared to the ancestor (GMI1000). This neighbour-joining phylogeny is based on single-nucleotide polymorphisms identified in the experimentally evolved strains. (A) and (B) M. pudica symbionts are in red and bold. Plant pathogens, opportunistic human pathogens and saprophytic strains are in blue, purple and black, respectively. Arrows indicate the acquisition of a symbiotic plasmid, either the ancestor of the C. taiwanensis one (pRaltaanc) during natural evolution or that of C. taiwanensis LMG19424 (pRaltaLMG19424) during experimental evolution. “
- Line 120. Ref 47 did actually show N fixation in the Agrobacterium transconjugants, on beans.
> The sentence was corrected (Lines 137-139) as “Indeed, the transfer of diverse rhizobium symbiotic plasmids into Agrobacterium tumefaciens or Escherichia coli, led to strains that were able to form either pseudo- or non-functional nodules [46, 47, 48] or nodules fixing low amounts of nitrogen [49].”